# Application of Elicitors in Two Ripening Periods of *Vitis vinifera* L. cv Monastrell: Influence on Anthocyanin Concentration of Grapes and Wines

**DOI:** 10.3390/molecules26061689

**Published:** 2021-03-17

**Authors:** Diego F. Paladines-Quezada, José I. Fernández-Fernández, Juan D. Moreno-Olivares, Juan A. Bleda-Sánchez, José C. Gómez-Martínez, José A. Martínez-Jiménez, Rocío Gil-Muñoz

**Affiliations:** Instituto Murciano de Investigación y Desarrollo Agrario y Alimentario (IMIDA), Ctra. La Alberca s/n, 30150 Murcia, Spain; dpq10f@carm.es (D.F.P.-Q.); josei.fernandez@carm.es (J.I.F.-F.); srjuanda@hotmail.es (J.D.M.-O.); juanantonio.bleda@carm.es (J.A.B.-S.); josec.gomez2@carm.es (J.C.G.-M.); josea.jimenez5@carm.es (J.A.M.-J.)

**Keywords:** phenolic compounds, methyl-jasmonate, benzothiadiazole, veraison, mid-ripening

## Abstract

In recent years, it has been demonstrated that the application of elicitors such as methyl-jasmonate (MeJ) and benzothiadiazole (BTH) to wine grapes can increase their phenolic and aromatic compounds if they are treated at the beginning of ripening (veraison). However, the veraison period is short, and it is not always possible to apply the treatments in a few days. Therefore, it would be of great interest to optimize the moment of elicitor application or extend the treatment period. The aim of this paper was to analyze during two consecutive years (2016–2017) the foliar application of MeJ, BTH, and a combination of both, during two different ripening periods of Monastrell grapes (veraison and mid-ripening), and determine the more appropriate moment to increase the concentration of anthocyanins. To carry out this aim, analysis of anthocyanins by HPLC in grapes and wines was mainly performed. The most suitable period for the application of MeJ, BTH, and MeJ + BTH was at mid-ripening, since the grapes showed a greater accumulation of anthocyanins at harvest. However, the MeJ + BTH treatment applied during veraison also obtained similar results, which would allow extending the application period if necessary. However, the increase in the anthocyanin content of grapes was not reflected in all the wines, which may have been due to reinforcement of the skin cell wall as a result of the application of elicitors. Further analysis is needed to improve the maceration process of the Monastrell grapes and the extraction of the anthocyanins that were increased by the treatments applied in the vineyard.

## 1. Introduction

There is growing interest in phenolic compounds as bioactive compounds because of their impact on human well-being, which is mainly influenced by their chemical structures [1]. Grapes are one of the most important sources of polyphenols for humans because their antioxidant and anti-inflammatory properties may contribute to preventing several diseases [2,3], as well as their consumption as fresh fruit or in red wines [3]. For both mentioned sources, secondary metabolites, particularly anthocyanins, come from the grape skin, where their concentration depends on the variety [4,5], production area [6], crop management techniques (irrigation, training system, pruning) [7,8,9] or the stress (biotic or abiotic) to which the vines are exposed [10].

Several studies have been carried out in this respect in wines with the Denominations of Origin “Jumilla”, “Yecla” and “Bullas” from Murcia (south-eastern Spain). *Vitis vinifera* L. cv Monastrell is the main grapevine in this region (more than 80% of the vineyards), and numerous projects have been undertaken to adapt its cultivation to new scenarios that include climate change [11], new consumer markets, and the obligation to attain environmentally respectful agriculture [12].

Other research studies have been developed to obtain new varieties from Monastrell with agronomic and oenological characteristics that are suited to this area [11], and different experiments have focused on increasing the phenolic composition of grapes through the use of elicitors [13,14,15,16,17]. Studies based initially on the defense against phytopathogens through the application of chemical elicitors/inducers have attracted much interest in viticulture [18], since their mechanisms activate the metabolic pathways of polyphenols related to defense mechanisms in the absence of a stimulus [19], and because they may represent an effective alternative to the use of conventional agrochemicals [20]. In recent years, it has been demonstrated that the application of elicitors such as methyl-jasmonate (MeJ) and benzothiadiazole (BTH) to wine grapes can increase their concentration of anthocyanins [15,16,17], stilbenes [21,22], and aromatic compounds [23], if they are treated at the onset of maturation (veraison). However, the veraison period is short, and it is not always possible to apply the treatments in a few days, so planning to use elicitors in large areas of vineyards or in unfavorable environmental conditions can become complicated. In this period, the grapes undergo critical physiological changes such as an increase in soluble solids, berry softening, and coloring [24], but these changes do not occur simultaneously and can vary widely among cultivars [25]. Therefore, to optimize the moment of elicitor application or extend the treatment period, it would be of great interest to determine the maturation stage (s) most suitable for the application of MeJ and/or BTH to favor the accumulation of anthocyanins in grapes.

Given the logistical and technological importance of the optimal moment of elicitor application, several studies have focused on the accumulation of anthocyanins, as well as color development; for example, after applying abscisic acid at different times during veraison to table grapes [24,26,27], grapes for juice production [28], or wine grapes [25,29]. Likewise, the evolution of the main phenolic compounds of Monastrell, Merlot, and Syrah wine grapes has been studied, in the period from the time of application (veraison) of MeJ and BTH to harvesting [30], and in all three varieties, treatment with elicitors increased the content of most of the studied phenolic compounds in grapes before harvest date, so it was proposed to optimize the moment of elicitor application in order to obtain their maximum effect at the moment of harvest, probably by delaying the application until some weeks after veraison. 

For the reasons mentioned above, the aim of this paper was to ascertain which of the two stages of ripening of Monastrell grapes (veraison and mid-ripening) was more advantageous for the foliar application of MeJ and BTH elicitors, or a combination of both, in order to determine the moment of application that increases the concentration of anthocyanins to the greater extent in grapes and consequently in wines.

## 2. Results and Discussion

### 2.1. Grapes

#### 2.1.1. Physicochemical Analysis in Grapes at Harvest

The results obtained for the different parameters measured are shown in Table 1. The results indicated that during the 2016 campaign, the average weight of 100 berries was 154.8 g, compared to the 165.7 g recorded in 2017. The higher berry weight observed in the latter year may have been due to a higher water absorption by the plant, due to the higher rainfall recorded in 2017 between July and August (54.2 mm), compared to the 2.6 mm accumulated during the same period in 2016 [31]. This fact may have accelerated berry growth through cell expansion [32], since during this stage, a great physiological activity occurs due to berry growth and ripening, accumulating a significant amount of water, and consequently increasing their weight. Reyero et al. (1999) [33] also observed that grapes of the Monastrell variety reached a larger size with increasing rainfall, as this variety responds quickly to water availability. It should be kept in mind that berry size is considered a quality factor, as the skin is the place where the main components that will be released during vinification are concentrated [34,35]. Therefore, a larger grape size will result in a lower skin to pulp ratio, which means that the phenolic and aromatic compounds will be more diluted in the must [36,37].

On the other hand, the effect of elicitor application on average berry weight during 2016 showed that treatments applied at mid-ripening had a greater effect on this parameter than treatments applied at veraison. Thus, we found that the MeJ + BTH treatment applied at veraison was the only one that managed to increase the average weight of the berries. In relation to this, other studies in which treatments were also carried out at veraison found that BTH applied to Monastrell increased berry weight only in one of the two study seasons [17]; the same happened with MeJ applied on Merlot and Cabernet Sauvignon varieties in a preliminary study, which also increased berry weight only in one of the two study seasons [38]. Likewise, increases in berry weight were observed with MeJ and BTH application in Syrah, while Merlot and Monastrell did not undergo changes [15]; however, in another trial, no increase was observed after MeJ application in Tempranillo [39,40]. On the other hand, we found that all treatments applied to the plants at mid-ripening (MeJ, BTH, and MeJ + BTH) caused increases in berry weight, BTH being the treatment that most increased this parameter.

In the 2017 season, Monastrell berries also responded differently to treatments depending on the time of application. Again, we observed that the response to treatments applied at mid-ripening on berry weight was more notable than treatments applied at veraison. In fact, at veraison, no differences were recorded between the average weight of treated and control grapes. However, at mid-ripening, the MeJ and BTH treatments did lead to increases in the average weight of the berries, thus obtaining results similar to those observed in the mid-ripening treatments in 2016. 

With regard to the soluble solids analyzed in the two seasons, the average concentration of the grapes in 2016 was 25.7 °Brix, which is a much higher value than the 22.5 °Brix recorded in 2017 (Table 1). In this sense, several factors could have influenced this variable, as bunch ripening can vary according to cultivar and environmental conditions [41]. It was suggested above that the increase in berry weight observed in 2017 may have been caused by an increase in water absorption due to higher rainfall during that year. This increase in berry weight may also have led to a dilution of the sugar content of the grapes, resulting in a lower soluble solids reading.

In addition to the higher rainfall recorded in 2017, there were more days with maximum temperatures above 30 °C in the period between the start of veraison and harvest, which amounted to 33 days in 2017 and 22 days in 2016. In addition, in 2017, the start of veraison was brought forward by 15 days, and the harvest date was 5 days earlier than in 2016, extending the ripening period of the Monastrell grapes by 10 days. However, despite this increase in the ripening period in 2017, the sugar concentration was lower at the time of harvest. In relation to this, it has been described that temperatures above 30 °C negatively influence the accumulation of sugars in the berries [42,43], as was the case in this trial.

In the 2016 season, the application of treatments did not cause statistically significant changes in the concentration of soluble solids of the treated grapes, if we compare them with the control grapes, both in the treatments that were applied at veraison and those applied at mid-ripening. However, we observed a slight increase in soluble solids, although not statistically significant, in the berries treated with BTH during mid-ripening. In this respect, the studies found regarding the effect of MeJ and BTH application on the concentration of sugars have shown varied results. Thus, it was found that the application of MeJ did not cause changes in the soluble solids of the Tempranillo variety [39,40,44] and Syrah [45]; however, it caused a reduction in soluble solids in the Merlot variety [38], Syrah [15], several clones of Monastrell [14], or in the Sangiovese variety [46], in which a 10-day delay in technological maturity was observed. 

Grapes treated with MeJ during veraison in the 2017 season also showed a lower soluble solids value than the control grapes. In the case of grapes treated at mid-ripening, BTH treatment produced grapes with higher sugar concentration than the control grapes, as observed in 2016, indicating that BTH treatment might advance the ripening process of Monastrell. However, other researchers observed a lower soluble solids concentration after BTH application in the Syrah variety [47]. 

Finally, the data for total acidity, tartaric acid, and malic acid showed no inter-annual differences, and no change was observed by the application of the treatments. However, the malic acid analysis of the grapes from the 2016 season revealed that the vines treated with BTH during veraison and MeJ at mid-ripening produced grapes with lower malic acid concentration than the control grapes. Regarding the effect of MeJ and BTH on pH, total acidity, tartaric acid, and malic acid, our results are in agreement with those obtained in previous research [14,16,48,49], where the authors also showed a slight or no influence of elicitor application on these parameters.

#### 2.1.2. Analysis of Anthocyanins in Grapes at Harvest by HPLC

The total anthocyanins presented in Figure 1 correspond to the sum of non-acylated and acylated anthocyanins. In this cultivar (Monastrell), non-acylated anthocyanins predominated (largest group), among which the individual anthocyanins 3 *O*-glucosides of delphinidin, cyanidin, petunidin, peonidin, and malvidin were identified (Table 2). In the case of acylated anthocyanins, Monastrell is a variety characterized by a low proportion of this group of anthocyanins [50]; however, 3-acetylglucosides and *trans-p*-coumaryl derivates of delphinidin, cyanidin, petunidin, peonidin, and malvidin were found; *cis*-*p*-coumaroyl and caffeoyl derivatives of malvidin were identified as well (Table 2). The anthocyanin profiles obtained for the different treatments were similar and reflect those found by other authors in the Monastrell variety [11,30,51].

Anthocyanin synthesis in Monastrell grapes during 2016 was higher than in 2017 for all the treatments (Figure 1A,B), suggesting that environmental conditions strongly affect the biosynthetic pathways of this variety, either by degradation and/or inhibition of anthocyanin synthesis [43,52], affecting their accumulation and distribution in berry skin [42,53]. In reference to this, it has been shown that inter-annual differences within the same variety can be much larger than inter-varietal differences during the same year [37,54]. 

As mentioned above, longer periods of temperatures exceeding 30 °C were recorded in the growing area during 2017, as was heavy rainfall (40 mm) a few days before the harvest [31]. These environmental conditions may have affected the accumulation and distribution of anthocyanins in the skin [53] and/or led to dilution of these polyphenols due to the greater absorption of water by the vine [17,55], as observed in the sugar concentration of the berries.

Moreover, the decision on the time of harvest was determined by the technological maturity, and not by the phenolic maturity, so it could be the case that the grapes were not harvested at the point of highest polyphenolic concentration of Monastrell. Related to this, a recent study analyzed the ripening evolution of Monastrell, Merlot, and Syrah grapes treated with MeJ and BTH during the veraison period [30], where they found that the maximum concentration of phenolic compounds was not always reached at the end of the ripening period but a few days before harvest.

During 2016, the application of all the treatments positively influenced the concentration of total anthocyanins with respect to the control (Figure 1A), and this increase in anthocyanin levels was statistically similar after the application of elicitors in both maturation stages (veraison and mid-ripening) studied. In 2016, no synergistic or antagonistic effect was observed when the elicitors were combined (MeJ + BTH).

In 2017, the treatments applied in mid-ripening were the most effective at increasing the concentration of total anthocyanins compared to treatments applied during veraison. However, the MeJ + BTH treatment applied in veraison that same year was the only one that managed to increase the concentration of total anthocyanins, pointing to a possible synergistic effect due to the combination of inductors. Few research studies have been reported on the synergism or antagonism between MeJ and BTH, although a cooperative interaction has been described in tomato [56] and in harvested table grape [57]. In addition, a recent study indicated that Monastrell grapes treated with MeJ + BTH had a higher anthocyanin content than control grapes [16]; however, the authors noted that the results were no better than a previous study [17], which involved the separate application of MeJ and BTH, at least proving the absence of antagonism between the two elicitors.

With respect to the individual treatments, several studies have shown that the exogenous application of MeJ can increase the concentration of anthocyanins in Monastrell grapes [14,16,17]; Syrah [45]; Tempranillo [39]; and Graciano wine grapes [44]. Likewise, BTH treatment in the Merlot variety also increased the concentration of anthocyanins [58]. However, all these trials with MeJ and BTH were only applied during the veraison period, resulting in it being difficult to make a comparative analysis with the results obtained in this trial. Related to this, although with a different type of elicitor, several trials have been described focusing on anthocyanin accumulation or skin color development by the application of abscisic acid before, during, and after veraison in grapes for juice production [28], in table grapes [24,26,27], or in wine grapes [25,29], all with satisfactory results. On the other hand, the application of abscisic acid 3–4 days after veraison in berries of the cultivar ‘Olympia’ significantly increased the anthocyanin content in the skins, two to three weeks after its application [59]. However, all these studies showed great variability in terms of the optimum time of application to achieve the maximum increase at harvest time. Therefore, up to date, it would be very difficult to establish the optimal application moment to apply a certain elicitor to a certain grape variety.

Non-acylated anthocyanins constituted about 80% of the total anthocyanins in the grape skins (Figure 1C,D). This high percentage obtained for this type of compound practically determined the results obtained in the concentration of total anthocyanins described above; that is, if the concentration of non-acylated anthocyanins is analyzed during the two campaigns, we observe that the increases recorded due to the treatments coincide with the increases recorded in total anthocyanins. 

The elicitors that increased the concentration of these compounds when applied in veraison 2016 were MeJ and BTH, while MeJ and MeJ + BTH increased them when applied at mid-ripening. In the treatments applied in 2017, only MeJ + BTH increased the concentration of non-acylated anthocyanins when applied during veraison, but all the treatments applied mid-ripening positively influenced the activation of these compounds. 

Additionally, acylated anthocyanins represented the remaining 20% of the total anthocyanins. In the 2016 season, all acylated anthocyanins increased in concentration following the application of the elicitors (Figure 1E) both at veraison and mid-ripening. However, in 2017, the concentration of acylated anthocyanins did not increase after the application of elicitors (Figure 1F); in fact, grapes treated with elicitors during the veraison stage showed lower concentrations than the control grapes, while the grapes treated in mid-ripening showed similar concentrations to the control grapes.

### 2.2. Wines

#### 2.2.1. Analysis of Anthocyanins in Wines by HPLC

As in grapes, the total anthocyanins analyzed in wines correspond to the sum of non-acylated and acylated anthocyanins (Figure 2). In this case, the non-acylated anthocyanins represented about 75% of total anthocyanins, and, besides the anthocyanins found in grapes, vitisins A and B were identified. Formation of the latter compounds occurs during alcoholic fermentation through the reaction of malvidin 3-*O*-glucoside with pyruvic acid (vitisin A) or with acetaldehyde (vitisin B) [39]. The anthocyanins analyzed are shown in detail in Table 3.

As expected, based on the results obtained in grapes, the wines made in the 2016 season showed a higher concentration of total anthocyanins than those of the 2017 season (Figure 2A,B). 

The concentration of total anthocyanins in wines produced during 2016 was only higher in the wines made from grapes treated with BTH at veraison; while wines from grapes treated with BTH and MeJ + BTH during mid-ripening had lower concentrations than the control wines. The results obtained in 2017 indicated once again that wines from grapes treated with BTH (veraison) had higher concentrations of total anthocyanins, as they did in 2016. Nevertheless, wines from grapes treated during mid-ripening showed slightly higher concentrations of total anthocyanins than control wines, but no statistically significant differences.

If total anthocyanins in grapes (Figure 1A,B) and total anthocyanins in the wines (Figure 2A,B) are compared, it can be seen that not all the grapes in which a higher concentration of total anthocyanins was observed following the application of elicitors produced wines with a higher concentration of anthocyanins than the control wines, as observed in previous trials [16,44,51]. It has been shown that knowledge of the amounts of anthocyanins in grape skins is not sufficient to estimate anthocyanin concentrations in wine, and this lack of correlation has been commonly attributed to the partial retention of these anthocyanins in the skin cells due to the barrier effect of the cell walls [60]. This phenomenon may be due to the fact that the application of elicitors has produced changes in the structure and composition of the skin cell wall, leading to reinforcement of the skin cell wall [38], thus hindering full anthocyanin extraction during the winemaking process.

Similar to total anthocyanins, wines from grapes treated with BTH (veraison) were the only ones that showed a higher concentration of non-acylated anthocyanins with respect to the control, during both the 2016 and 2017 seasons (Figure 2C,D). 

With regard to the concentration of acylated anthocyanins of wines produced in 2016 (Figure 2E), no statistically significant differences were found between the analyzed wines. However, in 2017 (Figure 2F), wines from grapes treated with BTH, both at veraison and mid-ripening, showed higher concentrations of acylated anthocyanins than the control wines. The importance of acylated anthocyanins in wine color should be emphasized, as they participate in the intramolecular copigmentation process, increasing and stabilizing the color of the wine [37]; thus, a higher concentration of acylated anthocyanins can translate into an increase in the sensory quality of the wines.

#### 2.2.2. Wine Color Parameters

The wines produced in 2016 had better color parameters than the wines produced in 2017 (Figure 3). In this respect, the color intensity (CI) is one of the most important parameters for determining a wine’s quality color being the first attribute perceived by consumers [61], and it is the anthocyanins that are largely responsible for this attribute [62]. The results obtained for the 2016 wines from grapes treated in veraison with MeJ, BTH, and MeJ + BTH had a higher CI than the control wines (Figure 3A), as did the wines from grapes treated with MeJ (mid-ripening). Only the wines made from grapes treated with BTH (veraison) showed higher CI for the 2017 season (Figure 3B). 

Therefore, most of the positive effect observed in the grapes due to the application of elicitors was reflected in the CI of the wines. In this sense, several studies have described that Monastrell grapes treated with MeJ and BTH during the veraison period resulted in wines with a higher CI than wines from control grapes [16,17,38,51]. The same trend was observed in trials with MeJ after application to Syrah varieties [45]; Merlot [38]; Graciano [44]; or Tempranillo [39].

When L* was measured in the elaborated wines (Figure 3C,D), those with a lower value than measured in the control wines were the same as those that presented a higher CI than the control wines mentioned above; that is, in 2016, wines from grapes treated at veraison with MeJ, BTH, and MeJ + BTH, as well as those treated with MeJ (mid-ripening). This was also the case for the 2017 wines made from grapes treated with BTH at veraison.

Lastly, 2016 wines made from grapes treated with MeJ (Veraison) and BTH (Veraison), as well as those treated with MeJ (Mid-ripening), showed lower hue than the control wines (Figure 3). On the other hand, hue values of all the wines made in 2017 were similar (Figure 3F).

Undoubtedly, the lower concentration of anthocyanins in the grape skins of the 2017 season had a direct impact on the color parameters of the wines produced, resulting in wines with lower CI and therefore with a higher L*.

### 2.3. Multivariate Discriminant Analysis

To classify the different individuals into alternative groups based on the analyzed variables (anthocyanins), a discriminant analysis was performed (Figure 4A,B). The generated figure visually showed the distinction between groups. The strong influence of the environmental conditions to which the vineyard was subjected was again confirmed. 

The statistical analysis performed on grapes clearly distinguished between the two years studied (Figure 4A), as well as between the times of treatment. Five discriminant functions were obtained, the first explaining 87% of the variance; the standardized coefficients that most influenced this discriminant function were cyanidin acetylglucoside and cyanidin 3-*O*-glucoside.

The statistical analysis performed on wine samples also allowed us to distinguish between the seasons studied and the moment of treatment; however, in 2016, the distinction observed between the control and the moments of application (veraison and mid-ripening) was more pronounced. Five discriminant functions were obtained, the first function explaining 92% of the variance, with coumaryl glucosides (petunidin and malvidin) having greater influence as standardized coefficients.

## 3. Materials and Methods

### 3.1. Experimental Design

The experiment, using the Monastrell variety, was carried out over two consecutive years (2016–2017) in Jumilla, Murcia (south-eastern Spain) (38°22′58.5′′ N, 1°26′30.8′′ W, elevation 380 m). Total precipitation and average temperature during the grape ripening period in 2016 were 2.6 mm and 24.9 °C, and 54.6 mm and 25.5 °C in 2017 [31]. The veraison and harvest dates in 2016 were August 5 and September 7, respectively. In 2017, the dates were July 21 and September 2, respectively. The study was performed on 14-year-old *Vitis vinifera* (syn. Mourvedre) red wine grapevines grafted onto 1103-Paulsen (clone 249) rootstock and trained in a vertical trellis system. Vine rows were arranged N-NW to S-SE with between-row and within-row spacing of 3 × 1.25 m. The soils are brown-limestone and stony, with low percentages of organic matter. The texture is predominantly silty-sandy, with high permeability and good aeration [63].

The experiments were conducted in a randomized block design, in which all treatments were applied to three replicates, using 10 vines for each replication. The protocol used to apply the different treatments, as well as the doses, have been described previously [38]. Plants (leaves and clusters) were sprayed with a water suspension of two elicitors: MeJ (methyl jasmonate) (Sigma Aldrich, St. Louis, MO, USA), at a concentration of 10 mM; BTH (benzo-(1,2,3)-thiadiazole-7-carbothioic acid S-methyl ester) (Sigma Aldrich, St. Louis, MO, USA) at a concentration of 0.3 mM or a mixture of both (at the same concentration of each component). Aqueous solutions (200 mL per plant) were prepared with Tween 80 (Sigma Aldrich, St. Louis, MO, USA) as wetting agent (0.1% *v/v*). Control plants were sprayed with aqueous solution of Tween 80 alone. The treatments were applied at two different stages of grape maturity: at veraison and at mid-ripening (3 weeks after veraison, since the Monastrell ripening period in this area lasts about 6 weeks). For all treatments, a second application was performed 7 days after the first in order to maximize anthocyanin accumulation based on the results of previous works with this and other grape cultivars [17,38]. When the grapes reached technological maturity (maximum sugar/acidity ratio, as determined from the control grapes), they were harvested and transported in boxes to the winery for physicochemical analysis and vinification.

### 3.2. Physicochemical Analysis of Grapes at Harvest

The traditional flesh parameters were measured: Total soluble solids (°Brix) using an Abbe-type refractometer (Atago RX-5000); titratable acidity using an automatic titrator (Metrohm, Herisau, Switzerland) with 0.1 N NaOH, and tartaric and malic acids using enzymatic kits from Boehringer 146 Mannheim GmbH (Mannheim, Germany). 

### 3.3. Vinifications

All vinifications were made in triplicate in 50-L stainless steel tanks using 50 kg of grapes, which were destemmed, crushed, and sulfited (8 g SO_2_/100 kg). Total acidity was corrected to 5.5 g/L with tartaric acid, and selected yeasts were added (Uvaferm VRB, Lallemand, 25 g/hL). A pre-fermentative cold maceration was carried out followed by traditional maceration. Tanks containing crushed grapes were placed in a cold room at 4 °C for 10 days, after which the tanks were returned to the winery to continue the traditional maceration process at 25 ± 1 °C. The fermentative pomace contact period was 10 days. Throughout the fermentation pomace contact period, the cap was punched down twice a day, and the temperature and must density were recorded. At the end of this period, wines were pressed at 1.5 bars in a 75 L tank membrane press. Free-run and press wines were combined and stored at room temperature. The analyses were carried out in triplicate at the end of alcoholic fermentation.

### 3.4. Analysis of Anthocyanins in Grapes and Wines

Grape and wine anthocyanins were determined according to Gil-Muñoz et al. [15]. Twenty berries for each treatment and replication were randomly taken from the different clusters. Grapes were peeled with a scalpel, and the skins were stored at −20 °C until analysis. Samples (2 g) were immersed in methanol (40 mL) in hermetically closed tubes and placed on a stirring plate at 150 rpm and 25 °C. After 4 h, the methanolic extracts were filtered through 0.45 μm nylon filters and directly analyzed by HPLC. The HPLC analyses were performed on a Waters 2690 liquid chromatograph (Waters, Milford, PA, USA), equipped with a Waters 996 diode array detector and a Licrochart RP-18 column (Merck, Darmstadt, Germany), 25 × 0.4 cm, 5-µm particle size, using water plus 5% formic acid (solvent A) and HPLC grade methanol (solvent B) as solvents at a flow rate of 1 mL min^−1^. Elution was performed with a gradient starting with 2% B to reach 32% B in 40 min, isocratic for 15 min, 50% B at 70 min, 60% B at 75 min, and then isocratic for 5 min. Chromatograms were recorded at 520 nm. Anthocyanins were quantified at 520 nm as malvidin 3-glucoside, using malvidin 3-glucoside chloride as external standard (Extrasynthèse, Genay, France). 

### 3.5. Wine Color Parameters

Samples were filtered and centrifuged, and a Shimadzu UV/visible spectrophotometer, model 1600PC (Shimadzu, Duisburg, Germany) was used to analyze color intensity (CI) and CIELAB parameters. CI was calculated as the sum of absorbance at 620 (blue component), 520 (red component), and 420 nm (yellow component) in undiluted wine [64]. The CIELAB parameters L* (lightness) and hue angle were determined by measuring the transmittance of the wine every 10 nm from 380 to 770 nm, using the D65 illuminant and a 10° observer angle [38].

### 3.6. Statistical Analysis

Significant differences among wines and grapes and for each variable were assessed by analysis of variance (ANOVA) using Statgraphics 5.0 Plus package (Statpoint Technologies, Inc., Warrenton, VA, USA). Duncan’s test was used to separate the means (*p* < 0.05) when the ANOVA test was significant, and a multivariate discriminant analysis was applied to identify the most discriminant variables.

## 4. Conclusions

Based on the results obtained in this trial, it is concluded that despite the fact that environmental conditions significantly influenced the physiology of the Monastrell grapes, the treatments MeJ + BTH applied in veraison, and MeJ, BTH, and MeJ + BTH applied in mid-ripening increased the concentration of total anthocyanins in the grapes in both the seasons studied, suggesting that mid-ripening is the most appropriate maturation period for applying these treatments. The treatment (MeJ + BTH) seems to be the most appropriate to increase the concentration of total anthocyanins, whether applied at veraison or mid-ripening since it would give more time to plan treatment, facilitating vineyard management even in unfavorable environmental conditions.

However, the increase in the anthocyanin content of grapes achieved by the above-mentioned treatments was not reflected in all the resulting wines, which was probably because the application of elicitors reinforces the skin cell wall, making it difficult to extract anthocyanins. Whatever the reason, it is clear that a more thorough analysis is necessary to improve the maceration process for Monastrell grapes and the extraction of the anthocyanins that were increased by the treatments applied in the vineyard. In this respect, this is the first attempt to optimize the time of application of these elicitors in order to maximize the accumulation of anthocyanins in grapes. The results provide information that will help understand the response of berries to treatments with MeJ and BTH in different circumstances.

## Figures and Tables

**Figure 1 molecules-26-01689-f001:**
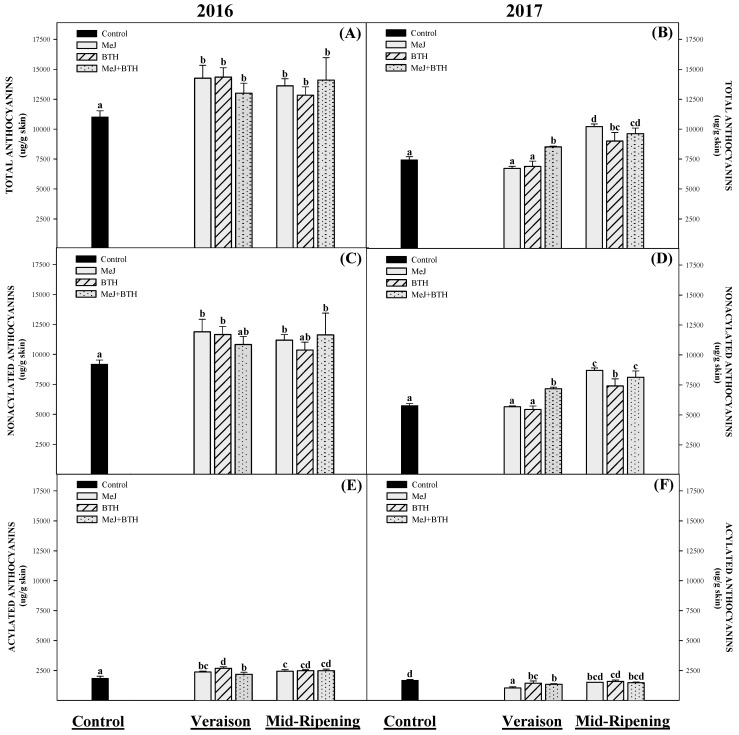
Concentration of anthocyanins measured by HPLC in Monastrell grape berry skin following treatment with MeJ, BTH, and MeJ + BTH applied on two different times: veraison and mid-ripening. Different letters indicate significant differences according to Duncan’s test (*p* < 0.05).

**Figure 2 molecules-26-01689-f002:**
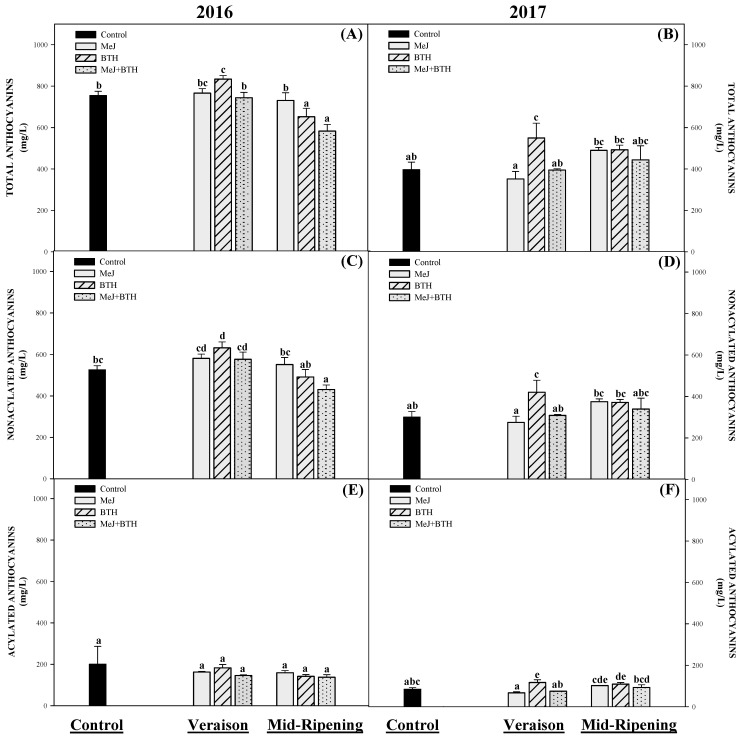
Concentration of anthocyanins measured by HPLC in Monastrell wines (expressed as mg/L) made from grapes treated with MeJ, BTH, or MeJ + BTH at two different times: veraison and mid-ripening. Different letters indicate significant differences according to Duncan’s test (*p* < 0.05).

**Figure 3 molecules-26-01689-f003:**
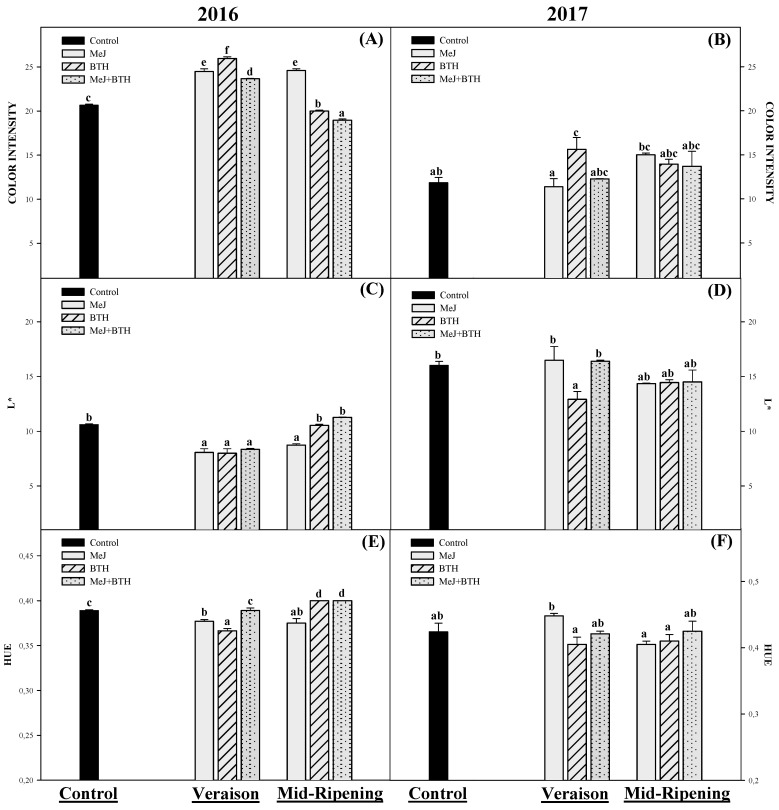
Color intensity, L* and hue in Monastrell wines made from grapes treated with MeJ, BTH, or MeJ + BTH at two different times: veraison and mid-ripening. Different letters indicate significant differences according to Duncan’s test (*p* < 0.05).

**Figure 4 molecules-26-01689-f004:**
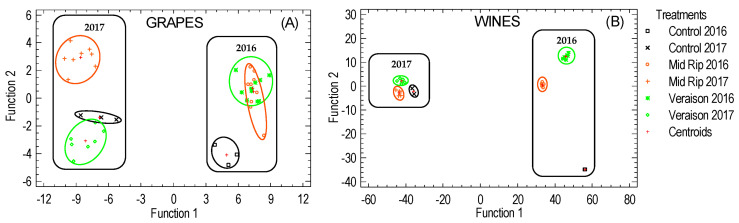
Distribution of the grape (**A**) and wine (**B**) samples in the coordinate system defined by the discriminants functions used to differentiate among treatments.

**Table 1 molecules-26-01689-t001:** Physicochemical characteristics of Monastrell grapes at harvest, treated with MeJ, BTH, and MeJ + BTH at two different application times: veraison and mid-ripening.

Year		Control	Veraison			Mid Ripening		
			MeJ	BTH	MeJ + BTH	MeJ	BTH	MeJ + BTH
	Weight 100 berries (g)	147.8 ± 6.2 a	146.7 ± 7.8 a	143.1 ± 10.7 a	162.9 ± 7.1 bc	156.1 ± 6.1 b	166.2 ± 9.5 c	160.6 ± 7.2 bc
	°Brix	25.9 ± 0.4 abc	25.2 ± 0.3 ab	25.6 ± 0.7 abc	26.1± 0.5 bc	25.0 ± 0.6 a	26.7 ± 0.6 c	25.5 ± 0.7 ab
	pH	3.9 ± 0.0 a	3.8 ± 0.0 a	3.8 ± 0.1 a	3.9 ± 0.0 a	3.8 ± 0.1 a	3.9 ± 0.1 a	3.9 ± 0.1 a
2016	Total acidity (g/L)	2.9 ± 0.4 a	2.8 ± 0.3 a	2.7 ± 0.1 a	2.7 ± 0.3 a	2.6 ± 0.4 a	2.8 ± 0.1 a	2.7 ± 0.1 a
	Tartaric acid (g/L)	4.1 ± 0.7 a	4.1 ± 0.8 a	3.8 ± 0.4 a	4.0 ± 0.5 a	3.8 ± 0.5 a	3.8 ±0.5 a	3.9 ± 0.3 a
	Malic acid (g/L)	1.5 ± 0.2 b	1.3 ± 0.0 ab	1.2 ± 0.1 a	1.5 ± 0.2 b	1.2 ± 0.1 a	1.5 ± 0.1 b	1.5 ± 0.1 b
	Weight 100 berries (g)	160.0 ± 7.2 a	157.4 ± 7.7 a	166.3 ± 10.3 a	166.7 ± 3.3 a	182.3 ± 0.8 b	178.0 ± 4.3 b	161.4 ± 3.1 a
	°Brix	22.6 ± 0.6 b	21.5 ± 0.9 a	22.8 ± 1.1 b	22.2 ± 0.4 ab	22.6 ± 0.5 b	24.2 ± 0.6 c	22.1 ± 0.6 ab
	pH	3.9 ± 0.1 c	3.8 ± 0.1 ab	3.9 ± 0.1 bc	3.7 ± 0.0 a	3.8 ± 0.1 abc	3.8 ± 0.1 abc	3.8 ± 0.1 ab
2017	Total acidity (g/L)	2.5 ± 0.4 a	3.1 ± 0.5 a	2.7 ± 0.4 a	3.1 ± 0.1 a	2.8 ± 0.4 a	2.8 ± 0.3 a	2.9 ± 0.3 a
	Tartaric acid (g/L)	3.7 ± 0.5 a	4.5 ± 1.5 a	3.9 ± 0.6 a	4.0 ± 0.7 a	4.0 ± 0.6 a	3.9 ± 0.5 a	3.9 ± 0.6 a
	Malic acid (g/L)	1.2 ± 0.0 ab	1.3 ± 0.1 bc	1.1 ± 0.1 a	1.3 ± 0.1 bc	1.2 ± 0.1 abc	1.3 ± 0.1 bc	1.2 ± 0.1 abc

Abbreviations: MeJ, Methyl Jasmonate; BTH, Benzothiadiazole; Data represent means ± SD. Different letters in the same row indicate significant differences according to Duncan’s test (*p* < 0.05).

**Table 2 molecules-26-01689-t002:** Concentration of anthocyanins measured by HPLC in the skin of Monastrell grapes treated with MeJ, BTH, and MeJ + BTH at two different application times: veraison and mid-ripening.

Year	Anthocyanins *(µg/g Skin)*	Control	Veraison			Mid Ripening		
			MeJ	BTH	MeJ + BTH	MeJ	BTH	MeJ + BTH
	Dp	1129 ± 161 a	1457 ± 255 b	1361 ± 34 b	1401 ± 57 b	1349 ± 51 ab	1274 ± 34 ab	1423 ± 110 b
	Cy	1271 ± 6 ab	1442 ± 19 abc	1296 ± 40 ab	1708 ± 21 bc	1425± 61 ab	1068 ± 225 a	1858 ± 612 c
	Pt	1276 ± 81 a	1667 ± 117 d	1609 ± 16 d	1492 ± 8 b	1551 ± 56 bc	1482 ± 33 b	1624 ± 69 d
	Pn	1387 ± 102 ab	1731 ± 6 ab	1673 ± 24 ab	1684 ± 17 ab	1610 ± 110 ab	1333 ± 291 a	1786 ± 492 b
	Mv	4108 ± 195 a	5579 ± 587 cd	5714 ± 171 d	4535 ± 367 ab	5247± 305 bcd	5204 ± 165 bcd	4932 ± 727 bc
	Dp Ac	31 ± 9 a	46 ± 4 b	42 ± 4 ab	44 ± 3 b	45 ± 8 b	51 ± 10 b	49 ± 8 b
	Cy Ac	30 ± 3 a	44 ± 3 bc	39 ± 4 ab	45 ± 3 bc	43 ± 4 bc	35 ± 7 ab	52 ± 13 c
	Pt Ac	54 ± 7 a	90 ± 9 c	85 ± 8 bc	75 ± 4 b	73 ± 4 b	80 ± 4 bc	75 ± 6 b
2016	Pn Ac	41 ± 5 a	59 ± 4 b	58 ± 9 b	52 ± 9 ab	54 ± 7 ab	49 ± 10 ab	62 ± 3 b
	Mv Ac	208 ± 22 a	311 ± 18 cd	315 ± 13 d	260 ± 20 b	286 ± 20 bcd	315 ± 18 d	275 ± 29 bc
	Mv Coum cis	27 ± 5 a	33 ± 3 bc	41 ± 3 d	30 ± 3 ab	34 ± 2 bc	34 ± 1 bc	37 ± 3 cd
	Dp Coum	113 ± 23 a	133 ± 9 b	161 ± 3 c	138 ± 8 b	151± 4 bc	143 ± 10 bc	146 ± 8 bc
	Mv Caf	26 ± 0 a	28 ± 4 ab	32 ± 2 ab	26 ± 1 a	32 ± 2 ab	32 ± 4 ab	34 ± 7 b
	Cy Coum	141 ± 2 a	149 ± 10 ab	164 ± 9 ab	174 ± 9 bc	157 ± 9 ab	143 ± 17 a	196 ± 35 c
	Pt Coum	196 ± 31 a	224 ± 23 ab	291 ± 4 e	228 ± 15 bc	262 ± 7 de	255 ± 14 bcd	259 ± 17 cde
	Pn Coum	163 ± 17 ab	160 ± 41 a	211 ± 34 ab	180 ± 26 ab	190± 16 ab	186 ± 27 ab	213 ± 23 b
	Mv Coum trans	808 ± 70 a	1105 ± 125 bc	1251 ± 39 c	932 ± 92 a	1115 ± 125 bc	1154 ± 25 bc	1078 ± 117 b
	Dp	759 ± 21 a	859 ± 120 ab	783 ± 56 a	1039 ± 49 d	1093 ± 10 d	986 ± 45 cd	912 ± 79 bc
	Cy	511 ± 42 a	1145 ± 270 b	573 ± 124 a	1120 ± 70 b	1925 ± 56 c	867 ± 93 ab	1964 ± 382 c
	Pt	889 ± 17 a	846 ± 120 a	885 ± 55 a	1061 ± 43 bc	1130 ± 24 c	1094 ± 66 bc	1009 ± 60 b
	Pn	567 ± 52 a	749 ± 137 b	539 ± 43 a	995 ± 1 b	1578 ± 45 d	1026 ± 98 c	1489 ± 101 c
	Mv	3048 ± 187 bc	2080 ± 246 a	2675 ± 295 b	2968 ± 106 b	2975 ± 73 b	3447 ± 315 c	2765 ± 81 b
	Dp Ac	29 ± 2 a	30 ± 2 a	31 ± 1 ab	32 ± 2 abc	37 ± 1 c	36 ± 4 bc	36 ± 1 bc
	Cy Ac	22 ± 1 a	33 ± 4 b	23 ± 1 a	35 ± 1 b	59 ± 1 c	35 ± 2 b	58 ± 7 c
	Pt Ac	53 ± 2 b	46 ± 8 a	52 ± 3 b	58 ± 1 cd	60 ± 3 de	66 ± 3 e	59 ± 2 d
2017	Pn Ac	24 ± 1 a	24 ± 3 a	23 ± 1 a	32 ± 0 b	54 ± 1 d	42 ± 2 c	51 ± 3 d
	Mv Ac	217 ± 15 cd	133 ± 14 a	176 ± 21 b	173 ± 2 b	194 ± 0 bc	225 ± 20 d	185 ± 18 b
	Mv Coum cis	34 ± 2 c	23 ± 2 a	31 ± 4 bc	25 ± 1 a	25 ± 1 a	27 ± 2 ab	25 ± 1 a
	Dp Coum	113 ± 2 c	82 ± 14 a	105 ± 10 bc	99 ± 9 bc	93 ± 2 ab	100 ± 6 bc	96 ± 4 ab
	Mv Caf	26 ± 6 a	18 ± 5 a	19 ± 1 a	17 ± 2 a	18 ± 1 a	25 ± 7 a	22 ± 6 a
	Cy Coum	89 ± 10 a	93 ± 8 ab	86 ± 8 a	113 ± 9 b	134 ± 2 c	107 ± 13 ab	137 ± 17 c
	Pt Coum	199 ± 6 d	113 ± 25 a	175 ± 24 cd	141 ± 9 b	154 ± 4 bc	166 ± 12 bc	153 ± 7 bc
	Pn Coum	94 ± 8 bc	68 ± 7 a	83 ± 10 b	102 ± 4 c	140 ± 3 e	127 ± 4 d	125 ± 3 d
	Mv Coum trans	769 ± 58 c	378 ± 50 a	627 ± 117 b	513 ± 26 b	545 ± 0 b	632 ± 53 b	534 ± 60 b

Abbreviations: MeJ, Methyl Jasmonate; BTH, Benzothiadiazole; Dp, delphinidin 3-*O*-glucoside; Cy, cyanidin 3-*O*-glucoside; Pt, petunidin 3-*O*-glucoside; Pn, peonidin 3-*O*-glucoside; Mv, malvidin 3-*O*-glucoside; Ac, acetylglucosides; Coum, coumarylglucosides; and Caf, caffeate glucoside. Data represent means ± SD. Different letters in the same row indicate significant differences according to Duncan’s test (*p* < 0.05).

**Table 3 molecules-26-01689-t003:** Concentration of anthocyanins expressed as mg/L at the end of alcoholic fermentation in Monastrell wines (2016 and 2017).

			Veraison			Mid-Ripening		
Year	Anthocyanins	Control	MeJ	BTH	MeJ + BTH	MeJ	BTH	MeJ + BTH
	Dp	46 ± 2 bc	54 ± 0 de	58 ± 1 ef	60 ± 0 f	50 ± 4 cd	42 ± 3 b	33 ± 2 a
	Cy	24 ± 1 bc	21 ± 2 ab	22 ± 2 abc	26 ± 0 c	32 ± 3 d	20 ± 2 ab	18 ± 1 a
	Pt	79 ± 1 b	89 ± 1 cd	95 ± 4 d	93 ± 4 d	81 ± 8 bc	71 ± 6 b	59 ± 3 a
	Pn	58 ± 1 ab	49 ± 7 a	54 ± 7 ab	54 ± 7 ab	64 ± 3 b	52 ± 1 ab	46 ± 3 a
	Mv	319 ± 4 bc	368 ± 16 de	403 ± 19 e	353 ± 5 cd	325 ± 16 bc	308 ± 24 ab	275 ±14 a
	Vitisin A	7.1 ± 3.0 b	5.3 ± 0.1 ab	4.8 ± 0.3 ab	5.9 ± 0.5 ab	5.3 ± 0.2 ab	4.7 ± 0.2 ab	3.9 ± 0.5 a
	Dp acetate	20 ± 10 a	13 ± 1 a	13 ± 1 a	14 ± 0 a	12 ± 1 a	11 ± 1 a	10 ± 2 a
	Vitisin B	8.8 ± 4.4 a	10.0 ± 3.7 a	8.2 ± 2.3 a	11 ± 3 a	8.6 ± 2.9 a	7.9 ± 2.3 a	7.1 ± 1.9 a
2016	Acetyl vitisin A	9.4 ± 8.9 a	6.9 ± 4.5 a	6.4 ± 4.2 a	4.6 ± 1.8 a	7.2 ± 5.2 a	5.0 ± 0.1 a	3.6 ± 1.2 a
	Cy acetate	12 ± 5 a	9.4 ± 0.2 a	9.0 ± 0.6 a	9.6 ± 0.7 a	9.2 ± 0.1 a	8.3 ± 0.1 a	7.3 ± 0.7 a
	Pt acetate	7.2 ± 5.9 a	6.5 ± 4.1 a	5.1 ± 2.6 a	6.6 ± 4.4 a	6.2 ± 0.8 a	4.8 ± 1.2 a	5.1 ± 2.8 a
	Pn acetate	15 ± 2 a	15 ± 1 a	17 ± 2 a	14 ± 1 a	15 ± 3 a	14 ± 2 a	13 ± 3 a
	Mv acetate + Dp coum	25 ± 6 a	25 ± 1 a	26 ± 1 a	23 ± 1 a	22 ± 1 a	22 ± 1 a	21 ± 2 a
	Pn caf	9.3 ± 4.0 a	7.6 ± 0.4 a	5.9 ± 0.1 a	7.4 ± 0.7 a	5.8 ± 0.2 a	6.1 ± 0.1 a	5.8 ± 0.6 a
	Cy caf + coum	11 ± 4 a	8.4 ± 0.5 a	9.3 ± 0.5 a	8.3 ± 0.9 a	10 ± 1 a	8.5 ± 0.2 a	7.7 ± 0.7 a
	Pt coum	15 ± 5 ab	14 ± 1 ab	18 ± 2 b	12 ± 0 ab	14 ± 2 ab	12 ± 1 ab	11 ± 2 a
	Mv coum cis	8.8 ± 2.0 b	4.0 ± 0.2 a	3.9 ± 0.9 a	4.2 ± 1.3 a	3.9 ± 0.0 a	2.9 ± 0.7 a	3.5 ± 0.1 a
	Pn coum	23 ± 5 b	10 ± 4 a	11 ± 2 a	7.9 ± 0.4 a	10 ± 1 a	10 ± 1 a	8.5 ± 1.6 a
	Mv coum trans	42 ± 0 ab	50 ± 1 c	65 ± 1 d	38 ± 3 a	50 ± 2 c	43 ± 4 ab	44 ± 3 bc
	Dp	22 ± 2 a	26 ± 1 ab	37 ± 3 c	31 ± 0 bc	34 ± 3 bc	33 ± 0 bc	29 ± 6 ab
	Cy	9.9 ± 0.1 a	16 ± 2 ab	17 ± 1 ab	19 ± 1 b	30 ± 2 c	17 ± 0 ab	24 ± 8 bc
	Pt	41 ± 4 a	41 ± 5 a	61 ± 10 b	46 ± 0 a	52 ± 2 ab	53 ± 1 ab	46 ± 8 a
	Pn	25 ± 1 a	28 ± 5 ab	42 ± 3 bc	33 ± 1 abc	55 ± 4 d	41 ± 0 bc	46 ± 12 cd
	Mv	203 ± 18 ab	164 ± 15 a	264 ± 36 c	179 ± 3 a	204 ± 3 ab	229 ± 13 bc	196 ± 18 ab
	Vitisin A	3.8 ± 1.0 a	3.6 ± 0.5 a	4.7 ± 0.3 a	4.1 ± 0.3 a	4.8 ± 0.7 a	3.7 ± 0.2 a	3.7 ± 1.1 a
	Dp acetate	6.2 ± 0.1 a	6.2 ± 1.0 a	8.7 ± 0.9 bc	6.9 ± 0.1 a	7.4 ± 1.1 ab	9.0 ± 0.4 c	7.2 ± 0.2 ab
	Vitisin B	1.9 ± 0.2 a	2.0 ± 0.1 a	2.2 ± 0.2 a	2.2 ± 0.0 a	2.4 ± 0.1 a	2.5 ± 0.2 a	2.3 ± 0.5 a
2017	Acetyl vitisin A	4.5 ± 0.3 a	4.4 ± 0.5 a	5.0 ± 0.1 ab	4.9 ± 0.0 ab	5.0 ± 0.2 a	5.7 ± 0.2 b	4.7 ± 0.9 a
	Cy acetate	4.1 ± 0.5 a	5.4 ± 1.8 a	9.8 ± 0.9 b	8.3 ± 0.1 b	11 ± 0 b	10 ± 0 b	9.4 ± 1.8 b
	Pt acetate	5.3 ± 0.4 a	4.9 ± 0.4 a	6.9 ± 0.0 b	5.3 ± 0.0 a	6.6 ± 0.2 b	6.8 ± 0.3 b	6.0 ± 0.9 ab
	Pn acetate	8.0 ± 0.8 ab	6.8 ± 1.0 a	11 ± 1 c	7.7 ± 0.2 ab	11 ± 0 c	11 ± 1 c	9.7 ± 1.8 bc
	Mv acetate + Dp coum	15 ± 1 ab	12 ± 3 a	18 ± 2 b	12 ± 0 a	16 ± 1 ab	17 ± 1 b	15 ± 1 ab
	Pn caf	3.8 ± 0.1 cd	2.7 ± 0.1 a	4.5 ± 0.3 e	3.0 ± 0.1 ab	3.6 ± 0.1 bc	4.2 ± 0.5 de	3.5 ± 0.1 bc
	Cy caf + coum	4.1 ± 0.0 a	3.8 ± 0.4 a	6.3 ± 0.6 c	4.5 ± 0.1 ab	7.1 ± 0.0 c	5.9 ± 0.0 c	5.7 ± 1.2 bc
	Pt coum	6.8 ± 0.7 bc	4.6 ± 0.2 a	9.3 ± 1.3 d	5.1 ± 0.2 ab	7.7 ± 0.3 cd	8.0 ± 0.7 cd	7.0 ± 1.3 bc
	Mv coum cis	2.9 ± 0.2 ab	2.2 ± 0.0 a	3.1 ± 0.3 b	2.1 ± 0.3 a	2.8 ± 0.2 ab	3.2 ± 0.7 b	2.8 ± 0.0 ab
	Pn coum	4.2 ± 0.0 a	3.0 ± 0.4 a	7.1 ± 0.6 c	3.6 ± 0.1 a	6.7 ± 0.2 bc	6.8 ± 0.0 bc	5.6 ± 1.2 b
	Mv coum trans	25 ± 3 a	15 ± 1 a	32 ± 4 d	16 ± 1 ab	23 ± 1 c	27 ± 3 cd	22 ± 3 bc

Abbreviations: MeJ, Methyl Jasmonate; BTH, Benzothiadiazole; Dp, delphinidin 3-*O*-glucoside; Cy, Cyanidin 3-*O*-glucoside; Pt, petunidin 3-*O*-glucoside; Pn, Peonidin 3-*O*-glucoside; Mv, Malvidin 3-*O*-glucoside; Caf, caffeate; Coum, coumarylglucosides. Data represent means ± SD. Different letters in the same row indicate significant differences according to Duncan’s test (*p* < 0.05).

## Data Availability

Data is contained within the article.

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
