# Peer review of "Application of Elicitors in Two Ripening Periods of Vitis vinifera L. cv Monastrell: Influence on Anthocyanin Concentration of Grapes and Wines"

_molecules, 2021, doi:10.3390/molecules26061689_

Round 1

Reviewer 1 Report

Thank you for submitting the manuscript “Application of Elicitors in Two Ripening Periods of Vitis vinifera L. cv Monastrell: Influence on Anthocyanin Concentration of Grapes and Wines” to Molecules.

The manuscript aimed to study the external application of two elicitor coumpounds for the growth of berries in grapes. The authors evaluated physicochemical characteristics, color, and anthocyanin content. In addition, the authors used principal component analysis to group the results obtained. In my opinion, the manuscript is well delineated and has interesting experimental results that justify its scientific publication. However, some points need to be improved.

1) In general, the abstract needs to be rewritten and give more value to the results obtained in the work. The authors have done much more analysis than appears in the abstract. A more complete abstract gives the reader a better idea of what he can find in the complete manuscript.

2) anthocyanins content: wouldn't it be better to express the value on a dry basis to isolate the effect of rain on the content of these biocomposites?

3) Figure 1, 2, and 3: I believe that graphics do not need the title "grapes" "wines" and "wine color" because the caption already describes what the product is about.

4) Figure 4: formatting is not compatible with the journal format.

Point-by-point

L18: the sentence is repeated in the abstract.

L23: Wasn't the best application time mid-ripening? This sentence is contradictory with the previous one.

L33: Change suggestion: “Grapes are one of the most important sources of polyphenols for humans because their antioxidant and anti-inflammatory properties may contribute to preventing several diseases [3,4], as well as their consumption as fresh fruit or in red wines [2].” In this case is necessary change the reference too.

L37, L42, and L62: there is an inconsistency in the use of the comma when the authors write a list:

Item A, Item B, or/and Item C

OR

Item A, Item B or/and Item C

Please correct all text.

L45: “different experiments have focused on increasing” where is the reference for these experiments? What proves this statement by the authors?

L47: cut the word “action”

L61: wine grapes are not grapes for processing? As a reader, this information is confusing.

L76-79: analysis of water content or total solids (by evaporation) of berries would confirm this statement. It is a simple, fast and practically free analysis. My suggestion is that the authors consider adding it to confirm the statement.

L102: °Brix is the unit of analysis. The correct thing is to write “soluble solids”. Please correct it throughout the text.

L106: without the moisture value of these berries all analyzes performed in this work will have depended on the rainfall value during that year. Once again I emphasize the importance of this analysis.

L150: Please improve the title of the table by giving more information about the experimental part of the manuscript. The table must be prepared for the reader regardless of the complete reading in the manuscript.

L417: how much solution was used per m² of vineyard plants? For the experiment to be replicated, it is necessary to indicate in addition to the concentration how much of the surface was covered by the solution.

Author Response

FIRST REVIEWER

English language and style

( ) Extensive editing of English language and style required
( ) Moderate English changes required
(x) English language and style are fine/minor spell check required
( ) I don't feel qualified to judge about the English language and style

Yes

Can be improved

Must be improved

Not applicable

Does the introduction provide sufficient background and include all relevant references?

( )

(x)

( )

( )

Is the research design appropriate?

(x)

( )

( )

( )

Are the methods adequately described?

( )

(x)

( )

( )

Are the results clearly presented?

( )

(x)

( )

( )

Are the conclusions supported by the results?

(x)

( )

( )

( )

Comments and Suggestions for Authors

Thank you for submitting the manuscript “Application of Elicitors in Two Ripening Periods of Vitis vinifera L. cv Monastrell: Influence on Anthocyanin Concentration of Grapes and Wines” to Molecules.

The manuscript aimed to study the external application of two elicitor coumpounds for the growth of berries in grapes. The authors evaluated physicochemical characteristics, color, and anthocyanin content. In addition, the authors used principal component analysis to group the results obtained. In my opinion, the manuscript is well delineated and has interesting experimental results that justify its scientific publication. However, some points need to be improved.

1) In general, the abstract needs to be rewritten and give more value to the results obtained in the work. The authors have done much more analysis than appears in the abstract. A more complete abstract gives the reader a better idea of what he can find in the complete manuscript.

Answer: Abstract has been rewritten and more information has been added.

2) anthocyanins content: wouldn't it be better to express the value on a dry basis to isolate the effect of rain on the content of these biocomposites?

Answer: We appreciate your suggestion, but unfortunately, we do not have any more samples available. It is a parameter that we usually present in this way, and we can only refer to fresh weight. But your suggestion is very interesting, and we will consider it for future trials.

3) Figure 1, 2, and 3: I believe that graphics do not need the title "grapes" "wines" and "wine color" because the caption already describes what the product is about.

Answer: Titles of figures 1, 2 and 3 have been deleted.

4) Figure 4: formatting is not compatible with the journal format.

Answer: Format of figure 4 has been changed.

Point-by-point

L18: the sentence is repeated in the abstract. Answer: Abstract has been rewritten and more information has been added.

L23: Wasn't the best application time mid-ripening? This sentence is contradictory with the previous one. Answer: Abstract has been rewritten and more information has been added.

L33: Change suggestion: “Grapes are one of the most important sources of polyphenols for humans because their antioxidant and anti-inflammatory properties may contribute to preventing several diseases [3,4], as well as their consumption as fresh fruit or in red wines [2].” In this case is necessary change the reference too. Answer: The entire sentence has been changed according to your suggestion, as well as the references. (Now in line: 37-40)

L37, L42, and L62: there is an inconsistency in the use of the comma when the authors write a list:

Item A, Item B, or/and Item C

OR

Item A, Item B or/and Item C

Please correct all text. Answer: This error has been reviewed and corrected throughout the document.

L45: “different experiments have focused on increasing” where is the reference for these experiments? What proves this statement by the authors? Answer: Several references have been added to support this statement. (Now in line: 51)

L47: cut the word “action” Answer: Word ¨action¨ has been eliminate. (Now in line: 53)

L61: wine grapes are not grapes for processing? As a reader, this information is confusing. Answer: Sentence has been corrected (Now in line: 67)

L76-79: analysis of water content or total solids (by evaporation) of berries would confirm this statement. It is a simple, fast and practically free analysis. My suggestion is that the authors consider adding it to confirm the statement. Answer: We appreciate your suggestion, but unfortunately, we do not have any more samples available. It is a parameter that we usually present in this way, but your suggestion is very interesting, and we will consider it for future trials.

L102: °Brix is the unit of analysis. The correct thing is to write “soluble solids”. Please correct it throughout the text. Answer: This error has been reviewed and corrected throughout the document.

L106: without the moisture value of these berries all analyzes performed in this work will have depended on the rainfall value during that year. Once again I emphasize the importance of this analysis. Answer: As mentioned above, unfortunately we do not have more samples available for this type of analysis. We can only refer to fresh weight results.

L150: Please improve the title of the table by giving more information about the experimental part of the manuscript. The table must be prepared for the reader regardless of the complete reading in the manuscript. Answer: The title of the table has been improved. (Now in line: 151-152)

L417: how much solution was used per m² of vineyard plants? For the experiment to be replicated, it is necessary to indicate in addition to the concentration how much of the surface was covered by the solution. Answer: In the experimental design we indicated that 200 mL/plant were applied, and that the treatment was applied to the entire plant (leaves and clusters). All this based on previous tests. But we do not determine the leaf area; so we cannot determine how much solution was used by m2 of vineyard plants. However, we appreciate your suggestion and will try to measure this parameter in subsequent trials.

Submission Date

17 February 2021

Date of this review

18 Feb 2021 01:42:26

Reviewer 2 Report

The paper is interesting, rich of data, the experimental design and the analytical methods are correct; moreover is has a practical meaning for growers and wine makers.

Some modifications/improvement are anyway needed for the paper to be published, as follows:

  • Line 31: the reference 1 is not appropriate; please change it and quote a paper related to the impact of phenolics on human health;
  • Line 33: the reference 2 is not appropriate; please change it and quote a paper dealing with human nutrition;
  • Line 34: the reference 4 is not appropriate, because it is not dealing with the role of polyphenols in preventing diseases;
  • Line 36: the reference 5 is correct but it is dealing with only intraspecific hybrids; please quote another one dealing with the majority of vinifera varieties.
  • Line 37: references 7 and 8 are not appropriate; please quote papers (reviews) dealing with cultural practices. Moreover “ breeding program” are not “crop management techniques” and therefore it has to be placed out of the  brackets;
  • Line 38: reference 9 is not appropriate. In the literatutre there are many reviews on this subject (for grapevine); please take the time to look for a proper reference.
  • Line 43: please add another reference, possibly in English;
  • Line 44: “…new varieties from Monastrell” and not “…of Monastrell”.
  • Line 47: reference 11 is not appropriate; please quote a review paper dealing with that issue;
  • Line 49: reference 12 is not appropriate;
  • Line 51: as concerning stilbenes and MeJ, more papers (other than Spanish) have been published earlier than the quoted ones; please look for them  and quote at least some of those;
  • Lines 55-56: The sentence is correct, but how is it related to the general reasoning?
  • Line 84: The two references are not appropriate because they do not describe the large array of skin components;
  • Line 108: Why “should be” if it is stated that 33 days in 2017 and 22 days in 2016 were above 30°C?
  • Lines 112-114: any record about the crop load per vine in 2016 and 2017? If there is a significant difference also this parameter might affect sugar concentration;
  • Line 176: it could be speculated (in order to explain the differences between the years) the lower day-night temperature gap in 2017 than in 2016;
  • Lines 412-413: please indicate the dates of the ripening periods (i.e. date of veraison and date of harvest);
  • Lines 414-415: what’s the meaning of “there-vine vertical trellis system”?
  • Lines 415-416: what’s the meaning of “(x 636.099; and 249.289) [29]”?
  • Line 416: please shortly describe the soil type;
  • Line 427: reference 4 is related to abscisic acid application and not to MeJ or BTH;
  • Lines 423-426: if I have well understood three sprays and not two were applied. Thesis 1: at veraison + 7 days later; thesis 2: at mid ripening; is it correct?
  • Line 483: as a rule, no references are reported in conclusions.
  •  

Author Response

REVIEWER 2

Open Review

English language and style

( ) Extensive editing of English language and style required
( ) Moderate English changes required
( ) English language and style are fine/minor spell check required
(x) I don't feel qualified to judge about the English language and style

Yes

Can be improved

Must be improved

Not applicable

Does the introduction provide sufficient background and include all relevant references?

( )

( )

(x)

( )

Is the research design appropriate?

(x)

( )

( )

( )

Are the methods adequately described?

(x)

( )

( )

( )

Are the results clearly presented?

( )

(x)

( )

( )

Are the conclusions supported by the results?

(x)

( )

( )

( )

Comments and Suggestions for Authors

The paper is interesting, rich of data, the experimental design and the analytical methods are correct; moreover is has a practical meaning for growers and wine makers.

Some modifications/improvement are anyway needed for the paper to be published, as follows:

  • Line 31: the reference 1 is not appropriate; please change it and quote a paper related to the impact of phenolics on human health; Answer: Reference has been changed. (Now in line: 37)
  • Line 33: the reference 2 is not appropriate; please change it and quote a paper dealing with human nutrition; Answer: Reference has been changed. (Now in line: 40)
  • Line 34: the reference 4 is not appropriate, because it is not dealing with the role of polyphenols in preventing diseases; Answer: Reference has been changed. (Now in line: 39)
  • Line 36: the reference 5 is correct but it is dealing with only intraspecific hybrids; please quote another one dealing with the majority of vinifera Answer: New references have been added. (Now in line: 42)
  • Line 37: references 7 and 8 are not appropriate; please quote papers (reviews) dealing with cultural practices. Moreover “ breeding program” are not “crop management techniques” and therefore it has to be placed out of the  brackets; Answer: Reference has been changed. And “ breeding program” has been removed. (Now in line: 42-43)
  • Line 38: reference 9 is not appropriate. In the literatutre there are many reviews on this subject (for grapevine); please take the time to look for a proper reference. Answer: Reference has been changed. (Now in line: 43)
  • Line 43: please add another reference, possibly in English; Answer: Reference has been changed. (Now in line: 43)
  • Line 44: “…new varieties from Monastrell” and not “…of Monastrell”. Answer: Paragraph has been corrected. (Now in line: 45)
  • Line 47: reference 11 is not appropriate; please quote a review paper dealing with that issue; Answer: Reference has been changed. (Now in line: 52)
  • Line 49: reference 12 is not appropriate; Answer: Reference has been changed. (Now in line: 55)
  • Line 51: as concerning stilbenes and MeJ, more papers (other than Spanish) have been published earlier than the quoted ones; please look for them and quote at least some of those; Answer: New references have been added. (Now in line: 57)
  • Lines 55-56: The sentence is correct, but how is it related to the general reasoning? Answer: We argue that most of the studies in which elicitors were applied were conducted at veraison, which is a relatively short period, and subject to great variability among cultivars. Therefore, it would also be interesting to extend the period of treatment application in order to favor the maximum accumulation of anthocyanins at harvest.
  • Line 84: The two references are not appropriate because they do not describe the large array of skin components; Answer: References have been changed. (Now in line: 92)
  • Line 108: Why “should be” if it is stated that 33 days in 2017 and 22 days in 2016 were above 30°C? Answer: Paragraph has been corrected. (Now in line: 117)
  • Lines 112-114: any record about the crop load per vine in 2016 and 2017? If there is a significant difference also this parameter might affect sugar concentration; Answer: Yes, there are records. The crop load was around 3.1-3.4 kg/vine, but no significant differences were found between the campaigns (data not shown). We thought it was more interesting to show the weight of 100 berries (Table 1).
  • Line 176: it could be speculated (in order to explain the differences between the years) the lower day-night temperature gap in 2017 than in 2016; Answer: We appreciate your suggestion, but unfortunately the weather stations in the area do not have detailed information on day and night temperatures; they only have: average temperatures, and average high and low temperatures. So we do not have enough information to support this suggestion.
  • Lines 412-413: please indicate the dates of the ripening periods (i.e. date of veraison and date of harvest); Answer: The veraison and harvest dates in 2016 were: August 5 and September 7, respectively. In 2017, the dates were July 21 and September 2, respectively. (Now in line: 415-417)
  • Lines 414-415: what’s the meaning of “there-vine vertical trellis system”? Answer: Paragraph has been corrected. (Now in line: 418)
  • Lines 415-416: what’s the meaning of “(x 636.099; and 249.289) [29]”? Answer: These are geographic coordinates. The information has been placed in a more appropriate place (Now in line: 414), and the units have been changed.
  • Line 416: please shortly describe the soil type; Answer: A new paragraph has been added. (Line : 419-420)
  • Line 427: reference 4 is related to abscisic acid application and not to MeJ or BTH; Answer: Reference have been changed. (Now in line: 431)
  • Lines 423-426: if I have well understood three sprays and not two were applied. Thesis 1: at veraison + 7 days later; thesis 2: at mid ripening; is it correct? Answer: No, that is not correct.

The treatments were applied at two different stages of grape maturity:

- At veraison (1st application) + 7 days later (2nd application).

- At mid-ripening (1st application) + 7 days later (2nd application).

  • Line 483: as a rule, no references are reported in conclusions. Answer: Reference has been removed. (Now in line: 487)

 Submission Date

17 February 2021

Date of this review

01 Mar 2021 11:35:17

Reviewer 3 Report

This paper provides some background information of the effect of  elicitors on anthocyanins content at different stages of application. While results show that there are significant changes in different anthocyanins content in grapes and wines. It could be more clearly if the following are added to the paper:

the relationship between the change of anthocyanins and quality of wine/grapes in terms of taste or commercial sense.

The relationship between the change of anthocyanins and the effect on health, e.g. antioxidant activities.

Author Response

REVIEWER 3

Open Review

English language and style

( ) Extensive editing of English language and style required
( ) Moderate English changes required
( ) English language and style are fine/minor spell check required
(x) I don't feel qualified to judge about the English language and style

Yes

Can be improved

Must be improved

Not applicable

Does the introduction provide sufficient background and include all relevant references?

( )

(x)

( )

( )

Is the research design appropriate?

(x)

( )

( )

( )

Are the methods adequately described?

(x)

( )

( )

( )

Are the results clearly presented?

(x)

( )

( )

( )

Are the conclusions supported by the results?

(x)

( )

( )

( )

Comments and Suggestions for Authors

This paper provides some background information of the effect of  elicitors on anthocyanins content at different stages of application. While results show that there are significant changes in different anthocyanins content in grapes and wines. It could be more clearly if the following are added to the paper:

 the relationship between the change of anthocyanins and quality of wine/grapes in terms of taste or commercial sense

Wines were subjected to a sensory discrimination test using a triangle test, to determine if there were organoleptic differences between the wines, and to check if they reduced or improved the sensory quality of the wines. But the results of these tests are included in another article, in which proanthocyanidins were analyzed (under review). Since we consider that both results were better complemented. However, the organoleptic characteristics of the wines produced were not altered to a greater extent by the treatments carried out in the vineyard.

The relationship between the change of anthocyanins and the effect on health, e.g. antioxidant activities.

With respect to relationship between the change of anthocyanins and the effect on health...... It is an interesting proposal, but unfortunately we do not perform this type of analysis, although we appreciate your suggestion, and we will try to add it to our analysis protocols for future trials.

Submission Date

17 February 2021

Date of this review

25 Feb 2021 09:28:14

Round 2

Reviewer 1 Report

Thank you for submitting the manuscript “Application of Elicitors in Two Ripening Periods of Vitis vinifera L. cv Monastrell: Influence on Anthocyanin Concentration of Grapes and Wines” to Molecules. The authors made the requested corrections and the manuscript improved a lot. In my opinion, the manuscript can be accepted for publication. 

Author Response

First of all, thank you very much for the time you have taken to review our manuscript. We appreciate all the suggestions and comments that he has made; they have certainly helped to improve its quality.